# An Individualized Approach of Multidisciplinary Heart Team for Myocardial Revascularization and Valvular Heart Disease—State of Art

**DOI:** 10.3390/jpm12050705

**Published:** 2022-04-28

**Authors:** Szymon Jonik, Michał Marchel, Zenon Huczek, Janusz Kochman, Radosław Wilimski, Mariusz Kuśmierczyk, Marcin Grabowski, Grzegorz Opolski, Tomasz Mazurek

**Affiliations:** 11st Department of Cardiology, Medical University of Warsaw, Banacha 1a Str., 01-267 Warsaw, Poland; michal.marchel@gmail.com (M.M.); zhuczek@wp.pl (Z.H.); jkochman@tlen.pl (J.K.); marcin.grabowski@wum.edu.pl (M.G.); grzegorz.opolski@wum.edu.pl (G.O.); tmazurek@kardia.edu.pl (T.M.); 2Department of Cardiac Surgery, Medical University of Warsaw, Banacha 1a Str., 01-267 Warsaw, Poland; rwilimski@gmail.com (R.W.); prof.kusmierczyk@gmail.com (M.K.)

**Keywords:** heart team, decision-making, coronary artery disease, mitral regurgitation, aortic stenosis, interventions

## Abstract

The multidisciplinary Heart Team (HT) remains the standard of care for highly-burdened patients with coronary artery disease (CAD) and valvular heart disease (VHD) and is widely adopted in the medical community and supported by European and American guidelines. An approach of highly-experienced specialists, taking into account numerous clinical factors, risk assessment, long-term prognosis and patients preferences seems to be the most rational option for individuals with. Some studies suggest that HT management may positively impact adherence to current recommendations and encourage the incorporation of patient preferences through the use of shared-decision making. Evidence from randomized-controlled trials are scarce and we still have to satisfy with observational studies. Furthermore, we still do not know how HT should cooperate, what goals are desired and most importantly, how HT decisions affect long-term outcomes and patient’s satisfaction. This review aimed to comprehensively discuss the available evidence establishing the role of HT for providing optimal care for patients with CAD and VHD. We believe that the need for research to recognize the HT definition and range of its functioning is an important issue for further exploration. Improved techniques of interventional cardiology, minimally-invasive surgeries and new drugs determine future perspectives of HT conceptualization, but also add new issues to the complexity of HT cooperation. Regardless of which direction HT has evolved, its concept should be continued and refined to improve healthcare standards.

## 1. Introduction

With an aging population, the increase in prevalence of atherosclerosis, coronary artery disease (CAD) and valvular heart diseases (VHD), degenerative and secondary to heart failure (HF) are expected. There is an unquestionable belief in the medical community that the standards of treatment should be incessantly improved with the use of experienced Heart Teams (HT) to provide the most satisfactory outcomes. An approach of multispecialist HTs is the most wanted to best assess the strengths and weaknesses of various treatment strategies for patients burdened with many co-morbidities. However, the HT management for “difficult” individuals is recommended in European and American guidelines both for myocardial revascularization and VHD [1,2,3,4,5,6,7]; this proceeding is mostly driven by expert opinion, whereas data from randomized-controlled trials (RCTs) supporting this approach is still scarce. Several studies suggest that through shared-decision making according to the guidelines, HT may improve overall outcomes. There is only a recognition, however, for real-life evidence we have to satisfy with observational studies describing outcomes of a multidisciplinary approach without a comparator.

The selection between interventional or medical treatment was originally based in the structure of HT from its first concept in the early 1980s [8,9]. We have known since then that each complex case should be discussed by at least three specialists: clinical and interventional cardiologists, and cardiac surgeons. With time and new treatment modalities, more specialists have been incorporated into HT structures and actively participated in the HT meetings depending on the complexity of the case. Moreover, there are many variables associated with the decision-making process as HT has been focused on the patient, not only selected disease treatment. Therefore, a holistic approach, risk assessment, specialists experience’ and the capabilities of the centre are also important (Figure 1).

Nowadays, the cooperation of HT seems to be virtually impossible without an experienced echocardiographist, radiologist and other imaging specialists who can assist with determination of disease severity, scope of the surgery and its expediency and feasibility. An anaesthesiologist can assess the perioperative risk for a patient who may undergo surgery or percutaneous procedure and give insights about the safety of general anaesthesia. A critical care intensivist is needed to guide the patient in the postoperative period. Furthermore, a nephrologist could help with those situations in which dialysis is contemplated or in case of complications such as acute kidney injury (AKI). A neurologist can assess the risk of cerebrovascular incidents and recommend prior intervention of the cerebral arteries when affected. Finally, a geriatrician could be involved in establishing of frailty status and purposefulness of interventional strategy. While the psychological aspect has been found to be an important factor in cardiovascular (CV) patients, a psychologist or physical therapist could also be incorporated into the decision process. Such approaches should be requested and actively implemented into HT protocols. The graphical cooperation of HT specialists is presented in Figure 2.

Below, we present, to our knowledge, the most extensive review of evidence from the literature supporting HT as desirable tool for optimal management of complex patients, providing improved outcomes and satisfactory quality of life.

## 2. Heart Team for Myocardial Revascularization

For many years, since CAD has been a leading problem of the modern world, ways to better diagnose and deal with this problem to improve patients’ management, prognosis and quality of life are being sought. With an increasing number of percutaneous and surgical options, new drugs improving symptoms and survival and improved collaboration between cardiologists and cardiac surgeons, an idea of HT has been implemented and still plays a leading concept in the real-life care of patients with CAD (class I recommendation in European and American guidelines) [1,2,3,4,5]. While choice of treatment for patients with acute conditions or less complex CAD may be single-minded, for individuals with stable multivessel disease (MVD), a HT consisting of non-invasive cardiologist, interventional cardiologist, cardiac surgeon and echocardiographist is considered the most wanted for selecting the optimal method of revascularization or disqualification for intervention. In the multicentre randomized Synergy Between PCI with Taxus and Cardiac Surgery (SYNTAX) trial, in which a local interventional cardiologist and cardiac surgeon at each site prospectively evaluated eligible patients with previously untreated left main (LM) disease and/or three-vessel disease (3-VD) to perform percutaneous coronary intervention (PCI) or coronary artery bypass grafting (CABG), an approach of HT for MVD patients was for the first time truly incorporated. The rates of major adverse cardiac or cerebrovascular events—MACCE (death from any cause, stroke, myocardial infarction (MI), or repeat revascularization) at 12 months—were significantly higher for PCI group (17.8%, vs. 12.4% for CABG; *p* = 0.002), mainly due to an increased rate of repeat revascularization (13.5% vs. 5.9%, *p* < 0.001). The rates of death and MI were similar between PCI and CABG, while the incidence of stroke was significantly more frequent in CABG group. The researchers reported that the use of antiplatelet drugs was high in the PCI arm (with 71.1% receiving a thienopyridine at 12 months). Additionally, both the use of statin, angiotensin-converting enzyme inhibitors or angiotensin II receptor blockers and calcium channel blockers was significantly higher after the study procedure in PCI group. Highly effective dual–antiplatelet and statin therapy may prevent thromboembolic incidents; hence, the lower rates of stroke in patients undergoing PCI. However, in this trial, the outcomes of participants disqualified from interventional treatment after HT evaluation and adherence to drug use in the optimal medical therapy (OMT) group were not assessed and this fact can be a kind of drawback [10]. After that, in 2014, Head SJ, et al., presented the final results of five-year follow-up for 1095 patients with 3-VD from SYNTAX trial randomly assigned to CABG (*n* = 549) or PCI (*n* = 546). The authors concluded that CABG should remain the standard of care resulted in significantly lower rates of death, MI, and repeat revascularization in CABG cohort with the rates of stroke independent of treatment strategy [11]. Furthermore, SYNTAX III Revolution trial identified computed tomography (CT) as non-invasive alternative to conventional angiography. This study did not focus on clinical endpoints and did not randomize patients but randomized physicians and surgeons from HT to make a decision on the best treatment for complex CAD. Two individual, blinded to each other HTs, composed of an interventional cardiologist, a cardiac surgeon, and a radiologist were randomized to evaluate CAD with either coronary computed tomography angiography (CTA) or conventional angiography in 223 patients with de novo LM stenosis or 3-VD. HT compliance in the assessment of patients’ qualification for PCI or CABG procedures (the primary endpoint) was very high at approximately 93%, whereas Cohen’s Kappa coefficient was 0.82, which indicates almost complete agreement between the two separate teams [12]. Additional insights from SYNTAX III Revolution revealed a secondary endpoint, including the physiological component using fractional flow reserve (FFR) derived from coronary CTA (FFR_CT_). It was demonstrated that coronary CTA evaluation with FFR_CT_ was feasible in 196 out of 223 MVD patients (87.9%). The inclusion of FFR_CT_ changed the HT treatment decision in 7% of the cases and modified the selection of vessels for revascularization in 12% as compared with a coronary CTA assessment alone. Moreover, FFR_CT_ reduced the proportion of patients with hemodynamically significant 3-VD from 92.3% to 78.8%, reclassifying them from intermediate and high to low SYNTAX score tertiles [13]. Very recently, the SYNTAX II strategy with assessment of both clinical and anatomical factors to guide myocardial revascularization was associated with improved 5-year clinical outcomes as compared with the SYNTAX trial, which evaluated anatomic variables only. For this study, 454 patients with de novo 3-VD were included and paralleled with 315 patients from the pre-defined SYNTAX PCI group and 334 patients from the pre-defined SYNTAX CABG cohort. The SYNTAX II strategy through functional assessment resulted in fewer lesions undergoing PCI, better optimization of PCI through the use of IVUS, more complete CTO revascularization, and optimal drug therapy. After 5 years, MACCE (all-cause death, any stroke, any MI, or any revascularization) occurred in 21.5% of SYNTAX II patients and was significantly lower than in the SYNTAX PCI cohort (36.4%, *p* <0.001). All MACCE components, except for stroke, were significantly less frequent in the SYNTAX II PCI group (*p* < 0.001). Also, the rate of in-stent thrombosis at 5 years was lower among SYNTAX II patients (1.4% versus 5.5%, *p* = 0.004). 

A similar rate of MACCE in the SYNTAX II group and the SYNTAX I CABG cohort were demonstrated (21.5% versus 24.6%, *p* = 0.35). In addition, optimized medical therapy was a part of SYNTAX II strategy. An increased use of statins at 5 years following revascularization (83% in SYNTAX-I PCI vs. 88% in SYNTAX II; *p* = 0.055) may be responsible for some of the improved outcomes of patients from the SYNTAX II cohort. Furthermore, pre-procedural loading dosing of statins may be associated with meaningful decrease in periprocedural rates of MI in SYNTAX II. Although the aspirin and dual antiplatelet therapy (DAPT) recommendations were significantly more frequent in the SYNTAX II group at discharge, rates of ADP antagonist prescription at 5 years were much higher in SYNTAX-I PCI. It is likely that this fact can be explained by lower rates of repeat revascularization/MI and utilization of new generation of drug-eluting stents (DES) with less dependence of DAPT in SYNTAX II cohort. Other CV medications were similarly used among SYNTAX II and SYNTAX-I PCI [14]. Afterwards, some observational studies evaluated HT approach for CAD-patients in single-center experiences. Bonzel T, et al., reported long-term outcomes of individuals with CAD qualified by HT to PCI. Out of 11,174 catheterizations for any reason 3408 catheterizations with a new diagnosis of CAD was analyzed by specialists to select optimal treatment modality and a total number of 1527 patients with first in life PCI for CAD were followed-up. The authors concluded that the multidisciplinary approach is a powerful tool for ad hoc and conference-based decision-making with desirable outcomes. 

During follow-up, CABG occurred in 15%, PCI in 37% and diagnostic catheterization in 65% of participants, while mortality of any course reached 51%. Mortality rates were similar in one-vessel disease (1-VD) and in patients matched for age and sex, but survival was significantly decreased in firstly-PCI patients with MVD [15]. Abdulrahman M, et al., presented the association between hierarchy in HT and recommendations for patients with isolated MVD. The decisions for CABG, PCI or OMT were made if the head of cardiovascular surgery (HOS) and the head of cardiology (HOC) were present, and only one of them was available or both directors were absent. When both HOC and HOS were present, only HOS was available, only HOC was available or both HOC and HOS were absent, the CABG-to-PCI ratios were 3.35, 4.88, 1.17 and 2.23, respectively.

This study demonstrated that HT decisions are not only related to current guidelines, but highly influenced by hierarchy among the members of the HT [16]. Another study assessed the long-term survival of 366 patients (74.1% with MVD, mean age 69 ± 11 years) consulted at 51 HT meetings. Depending on the baseline clinical characteristics and risk assessment, patients were qualified for CABG+OMT (*n* = 102), PCI+OMT (*n* = 127) or OMT only (*n* = 137). Also, the multinomial logistic regression analysis was performed to define factors associated with HT strategy, which revealed that patients had increased odds of receiving PCI if they were in cardiogenic shock or had 3-VD (not including left main stenosis (LMS)), CABG was recommended for younger and with isolated LMS, while OMT for the oldest and with diabetes mellitus (DM). 3-year survival was 60.8%, 84.3% and 90.2% in the OMT, OMT+PCI and OMT+CABG cohorts, respectively. For patients who underwent HT discussion and implementation of any revascularization strategy, no significant difference in mortality between CABG and PCI cohorts was demonstrated [17]. In 2019, Dominiques, et al., presented HT management for nearly 1000 patients with CAD, 69.4%, simple CAD and 30.6% for MVD qualified after careful HT evaluation to CABG, PCI, OMT or additional diagnostic methods depending on the number of affected coronary vessels, HT decisions and patients’ preferences and followed with median (interquartile range (IQR)) time of 4.6 (4.2–5.0) years. 

The authors reported no association between proximal left anterior descending (LAD) involvement and all-cause death for patients with 1-VD or 2-VD (16.4% vs. 15.7% for non-proximal LAD, *p* = 0.70), while for individuals with complex CAD the overall mortality was significantly increased in LMS with 2-VD or 3-VD (26.9%), *p* = 0.019 [18]. Young, et al., reported prospectively evaluated data of 166 high-risk patients with CAD qualified to CABG (*n* = 49), PCI (*n* = 79), OMT (*n* = 34) or hybrid therapy (*n* = 1) following HT decisions. The median (IQR) number of physicians per HT council was 6 (5–8). With increasing Society of Thoracic Surgeons Predicted Risk of Mortality (STS-PROM: low, intermediate, high) operative risk, CABG was performed less often and OMT was recommended increasingly, while no trends in HT decisions for CABG, PCI or OMT by SYNTAX score tertiles were observed. Among 129 patients who underwent revascularization (CABG or PCI) in-hospital and 30-days mortality was 3.9% and 4.8%, respectively, while the 30-day unplanned rehospitalization rate was 16.4%, 22.4% and 17.6% for PCI, CABG and OMT-patients, respectively [19]. Another study compared HT decisions and delay to revascularization for MVD-patients from 2 groups: evaluating by HT (93) and control group (93) matched according to clinical and angiographic characteristics. No significant differences in CV risk, left ventricular (LV) dysfunction, STS and SYNTAX scores between these two groups were observed. After HT discussion, the percentage of patients qualified to CABG resulted in 63% and was significantly higher than in control group −23% (*p* < 0.01). HT management led to a significant delay to PCI, while delay to CABG was not affected [20]. Tsang MB, et al., reported very interesting results from a study of 234 patients with MVD comparing the treatment originally implemented by interventional cardiologists (2012–2014) with recommendations proposed by members of 8 blinded HTs (2017–2018). Between the original decisions of the interventional cardiologists and the results of the HT consultations, a different decisions occurred in nearly one-third of the cases. HT members indicated statistically insignificant, but numeric bias toward the procedure of their specialty. 

Overall, as the choice of the treatment strategy is regarding, there were no statistically significant differences between interventional cardiologists and HT members for CABG (*p* = 0.62) or PCI (*p* = 0.15), while OMT was less frequently recommended originally by interventional cardiologist than by the HT members (*p* = 0.04). ([21]; and comment—[22]). We also served our internal single-centre experience with mean (standard deviation (SD)) follow-up of 37 (14) months for 1286 participants with severe CAD (3-VD and/or LM disease) and fully implemented HT decisions (OMT, CABG or PCI for 251, 356 and 679 patients, respectively). The ratio of primary endpoint—MACCE (overall death, stroke, MI, or repeat/need for revascularization) was significantly increased in OMT-group as compared with CABG or PCI (*p* < 0.05), while considering interventional strategies only—CABG was associated with reduced rates of MACCE and repeat revascularization, while the superiority of PCI for stroke and disabling stroke was observed (*p* < 0.05). 

The general health status assessed at the end of follow-up was significantly more satisfactory for patients who underwent revascularization than in OMT-group (*p* < 0.05) [23]. Current evidence summarizing the role of HT for treating patients with CAD is presented in Table 1.

## 3. Heart Team for Aortic Valve Stenosis

Over the years we have observed an improved level of health care and we predict a further increase in life expectancy, and the prevalence of degenerative aortic stenosis (AS) due to aging of the population is also expected. The problem is urgent as AS is the most widespread VHD in the world and still remains the most common indication for valve intervention in Europe and North America [6,7]. The surgical aortic valve replacement (SAVR) has previously been the standard of care for AS-patients, improving both symptoms and prognosis, while since 2007, the less invasive transcatheter aortic valve replacement (TAVR) has been commercially available. Currently, the state of the art for the treatment of patients with symptomatic AS includes both conventional surgery, percutaneous treatment (TAVR) and conservative approach—OMT, depending on many variables. Although many RCTs have compared outcomes of patients with high-, intermediate- and low- risk AS who were treated with SAVR or TAVR, the role of HT was poorly underlined in these studies [24,25,26,27,28,29,30]. Admittedly, HT was used to evaluate the baseline status of patients and determine the perioperative risk, but not as decision-making tool for selection of the optimal treatment modalities. Current recommendations for intervention in AS–patients are guided by the RCT findings and compatible with real-world HT cooperation for individual patients (many of whom not meet the RCT inclusion criteria) [6,7]. 

AS is a very heterogeneous condition and the most beneficial procedure should be carefully considered by the individual HT, accounting for age, life expectancy, comorbidities and frailty, anatomical and procedural characteristics, prosthetic heart valve durability, feasibility of vascular access and local experience with long-term outcomes. While waiting for an RCT assessing the efficacy of HT approach for AS-patients, we have to be content with data from observational studies only. 

For the first time, Dubois, et al., demonstrated prospective management of 163 high-risk patients with AS qualified after HT evaluation to transcatheter aortic valve implantation (TAVI)—73, SAVR (35) and OMT with or without percutaneous transluminal aortic valvuloplasty (PTAV)—55 patients. The authors reported that TAVI and SAVR was found to be significantly superior to OMT/PTAV for all-cause mortality and CV death and nonsignificantly superior to OMT/PTAV for repeat hospitalizations for CV cause at 1 year. For interventional procedures, the combined safety endpoint (overall mortality, major stroke, life-threatening bleeding, AKI stage 3, periprocedural MI, major vascular complication or repeat procedure for valve-related dysfunction) at 30 days favored TAVI, while the combined efficacy endpoint (overall mortality after discharge, rehospitalization for CV causes and prosthetic heart-valve dysfunction) at 1 year supported AVR approach [31]. Similarly, Thyregod HGH, et al., reported very poor prognosis for patients with severe AS qualified by HT to OMT with survival rate significantly lower as compared with TAVI– and SAVR–patients when using Cox regression analysis adjusted for age and gender (*p* < 0.01). The HT proposed intervention in 93% of patients with severe AS despite high age, advanced symptoms and a high burden of co-morbidity, while those for whom HT did not propose to undergo any intervention were older, had a higher prevalence of chronic obstructive pulmonary disease (COPD), peripheral artery disease (PAD), previous MI and cerebrovascular disease. Disqualification from any procedure resulted in a very dismal prognosis in OMT-cohort with only 57 and 26% surviving to 1 and 3 years, respectively [32]. 

Data from the Belgian centre revealed that TAVI as carefully discussed and passed by HT translates into similar outcomes and shorter hospital stay as compared with SAVR even for higher-risk patients. Bakelants E, et al., presented the cooperation of HT in a health-economic context with limited accessibility for transcatheter procedures. For 405 prospectively observed high-risk patients with AS qualified for SAVR—98, TAVI—188 and OMT/PTAV—116, TAVI and SAVR was found to be significantly superior to OMT/PTAV for all-cause mortality and CV death at 1 year, while no differences in stroke/transient ischemic attack (TIA) and CV-rehospitalization between groups after 30 days and at 1 year were observed [33]. 

In a retrospective study by Rea CW, et al., 243 individuals with severe AS were assessed by HT and qualified to SAVR—26, TAVI—200 and OMT—17. No significant differences in age or perioperative risk assessed by EuroSCORE II between these three groups were observed. The authors reported that survival outcomes after TAVI and SAVR were similar with each other (93% vs. 84% for SAVR at 1 year and 85% vs. 84% for SAVR at 2 years) and similar to the age-matched general population with both being longer than for patients receiving only OMT (73% and 54% at 1– and 2–years, respectively, *p* = 0.002) [34]. A total number of 286 high-risk patients with AS discussed by HT and qualified for SAVR (*n* = 53), TAVR (*n* = 210) and OMT (*n* = 23) were prospectively evaluated with median (IQR) follow-up of 18 (11–26) months in the study of Tirado–Conte G, et al. The authors reported an increasing number of patients referred for HT discussion, with a 26% growth between study periods. Importantly, 20% of patients in the SAVR-cohort underwent a concomitant valve intervention. In-hospital mortality was 7.5% for SAVR, compared with 3.4% in the transfemoral TAVR group (*p* = 0.447). 1– and 2–year all-cause mortality did not significantly differ between SAVR and TAVR groups (14.0% vs. 17.2% for TAVR at 1 year and 17.2% vs. 25.9% at 2 years), while patients referred to OMT had the worst prognosis with only one-third survived 1– and 2–years [35]. 

In our retrospective study, we evaluated patients presented to our internal HT during a period of 4 years. Finally, 482 participants with severe AS and completely implemented HT decisions (OMT, TAVR and SAVR for 79, 318 and 85, respectively) were included and followed for adverse events with a period of about 2.5 years. SAVR and TAVR were found to be superior to OMT for primary (all-cause mortality, non-fatal disabling strokes and non-fatal rehospitalizations for AS) and all secondary endpoints (*p* < 0.05). Comparing interventional strategies only, TAVR was associated with a reduced risk of AKI, new onset of atrial fibrillation and major bleeding, while the superiority of SAVR for major vascular complications and need for permanent pacemaker implantation was observed (*p* < 0.05). The quality of life assessed at the end of follow-up was significantly better for patients who underwent TAVR or SAVR than in OMT-group (*p* < 0.05) [36]. Current evidence from observational studies summarizing the role of HT for treating patients with AS was presented in Table 2.

## 4. Heart Team for Mitral Regurgitation

The current evidence demonstrating prognosis of MR-patients treated surgically, percutaneously or with OMT is still scarce, and although multiple reports have published survival data, only a few have compared outcomes post interventional approaches or OMT. So far, two RCTs: EVEREST II—Endovascular Valve Edge-to-Edge Repair Study [37] and COAPT—Cardiovascular Outcomes Assessment of the MitraClip Percutaneous Therapy [38] has reported results of severe MR treatment. In the EVEREST II trial 279 patients with moderately severe or severe MR (grade 3+ or 4+) were randomly assigned to receive MitraClip (MC) or mitral valve (MV) surgery—repair or replacement in a 2:1 ratio. Although percutaneous repair was less effective at reducing MR than conventional surgery and patients from surgery cohort had significantly better outcomes at 12 months (primary endpoint—freedom from death, surgery for MV dysfunction, and from grade 3/4+ MR)—73% vs. 55% in MC-group, *p* = 0.007, both groups achieved similar improvements in clinical outcomes [37]. At 5 years, the rate of the composite endpoint of freedom from death, surgery for residual MR, or 3/4+ MR in the intention-to-treat population was 44.2% vs. 64.3% in the MC and surgical groups, respectively (*p* = 0.01), however, mortality rates did not favor surgical approach (20.8% vs. 26.8% for surgery, *p* = 0.4) [39]. In the COAPT study, 610 patients with HF and moderate—to–severe (3+) or severe (4+) secondary MR who remained symptomatic despite maximally-tolerated OMT were randomized in a ratio 1:1 to receive MC with OMT or OMT only. At 24 months, MC with OMT approach as compared with OMT alone was associated with significantly improved outcomes: the annualized rate of all hospitalizations for HF (35.8% vs. 67.9%, respectively, *p* < 0.001) and overall mortality (29.1% vs. 46.1%, respectively, *p* < 0.001). The rate of freedom from device–related complications at 12 months was very high—96.6% (*p* < 0.001 for comparison with the performance goal) [38]. However, in these randomized studies, the involvement of HT for optimal decision-making for patients with symptomatic MR was not detailed. Although an approach of experienced HT was emphasized in current guidelines for VHD [6,7], the position of HT in the treatment of MR-patients is based only on experts’ opinion and data from some observational studies. In the study by Heuts S, et al., 158 patients with MR were qualified after HT discussion to different treatment strategies—surgery (isolated or concomitant mitral valve replacement (MVR)—67 patients), transcatheter intervention (MC or mitral valve repair (MVP)—20 patients) or OMT (71 patients). 30-days mortality were 3 (4.4 %), 0 (0.0 %) and 3 (4.2 %) for surgery, MC/MVP and OMT, respectively. Using statistical analysis with a median follow-up of 450 days for the various treatment options, an improved survival for surgically treated patients was revealed [40]. 

In another research, Külling M, et al., presented observational single-center report of 400 patients managed for MR. Followed by HT decisions, 179 patients (44.8%) were treated using MC, 185 (46.2%) by MVP and 36 (9.0%) by MVR. Outcomes with mean follow-up (SD) time of 32.2 (17.6) months favored patients treated with MVP who had higher 4-year survival (HR 0.40 (95% CI 0.26 to 0.63), *p* < 0.001) and fewer combined endpoints (all-cause mortality, cardiovascular (CV) rehospitalization and MV reintervention) as compared with MVR and MC groups [41]. Very recently, Nia PS., et al., reported that dedicated mitral HT provide improved care for patients with MV disease. A total number of 1145 patients—641 managed by the dedicated mitral HT and 504 by the general HT were observed for adverse events. At 1 year, the mortality was 74 (14.7%) for the general HT as compared with only 57 (8.9%) for the dedicated mitral HT (*p* = 0.002). At 5 years, survival probability was measured as 0.74 for the dedicated HT as compared to 0.70 for the general HT (*p* = 0.04). 

The limitation of this study could be its non-randomized character; however, this kind of approach seems to be not necessary as it is intuitively obvious that specialists provide better management than generalists [42]. We also reported our plot in this topic providing outcomes and quality of life of patients with severe MR consulted by our internal HT. With mean (SD) follow-up of 29 (15) months 157 individuals with severe MR and completely implemented HT decisions (OMT, MC or MVR for 53, 58 and 46 patients, respectively) were included. MVR and MC were significantly superior to OMT for primary endpoint (CV death) and all secondary endpoints—overall mortality, non-fatal MI, non-fatal strokes, non-fatal hospitalizations for HF exacerbation and any CV events (*p* < 0.05). However, for interventional strategy—no significant differences between MVR and MC approach were observed. At the end of follow-up, physical, mental and total qualities of life for all alive participants were significantly improved for MVR-patients, then for MC and the poorest in OMT-group [43]. Current evidence from observational studies summarizing the role of HT for treating patients with MR is presented in Table 3.

## 5. Limitations

As we noted in the introduction, currently the main limitation of the HT concept still remains the lack of well-founded, step by step-planned RCTs comparing the long-term outcomes of patients treated with and without the HT approach. To date, evidence of the advantages of implementing multidisciplinary decision-making has been derived mainly from expert opinion and observational studies without a comparator. Unfortunately, in most of the large studies we referenced in this manuscript, such as SYNTAX, SYNTAX II, PARTNER, PARTNER 1, NOTION, PARTNER 2, SURTAVI, EVEREST II or COAPT [10,14,24,27,28,29,30,37,38], the main theme is head-to-head comparison of various treatment options for myocardial revascularization, AS or MR, rather than an importance or specific role of HT in the management of patients with these diseases. Therefore, the performing of well-designed RCTs with hard clinical endpoints remains one of the most important perspectives for a future HT concept. Additionally, for patients with CAD and VHD, adherence to physician recommendations and regular drugs usage is very important factors of future prognosis and quality of life. Unfortunately, these parameters are very difficult to measure and often remain a matter of mutual trust between the doctor and the patient. Also, among the articles regarding HT that we have cited in this manuscript, this issue is overlooked and very rarely raised [10,14]. So, a proper qualitative assessment of adherence to medical recommendations is still an unexplored issue and adds to the limitations of our review.

## 6. Future Perspectives

In this review, we presented the most extensive summary of the established and emerging evidence for the role of HT for myocardial revascularization and VHD—predominantly AS and MR as randomized or at least observational studies concerning management of HT for other heart valve defects are still unavailable in the literature. We have described HT (1) as recommended by guidelines for selection of optimal treatment modalities for complex patients, (2) emphasized that RCTs are desirable for future evaluation of HT concept (although proper design of such studies will be difficult), (3) highlighted importance of HT for perioperative risk assessment, and (4) proved that HT through weighing-up of the risks and benefits of each strategy for individual patient may provide improved outcomes in real-life clinical practice. Nowadays, the COVID-19 pandemic has brought telemedicine to the forefront of medical care and we assume that in the future, the HT meetings will also be digital. The remote patient’s management with HT aided by artificial intelligence may be the next step of the development of HT concept. 

At this point, we need to underline that independently of future directions of HT, patient’ preference should always be on first place and shared-decision making could ameliorate balancing between mortality benefit and other patients–related matters such as periprocedural complications, the length of in-hospital stay and quality of life. 

However, knowledge alone is not sufficient for patients to feel comfortable stating their own preferences; rather, a clear invitation from specialists for shared-decision making must be expressed. The future concept of HT should be developed with optimization of PCI procedures (including functional assessment, intravascular ultrasound and improved techniques of chronic total occlusion management), minimally invasive valvular surgeries and using new drugs improving symptoms and survival. The improvement of HT collaboration (members’ interactions, feedback algorithms and patient involvement in decision steps) and subsequent RCTs would increase the HT importance and its implication in real-life clinical conditions. 

Despite all future HT evolutions, one should be constant: the patient should remain at the main centre of each HT.

## Figures and Tables

**Figure 1 jpm-12-00705-f001:**
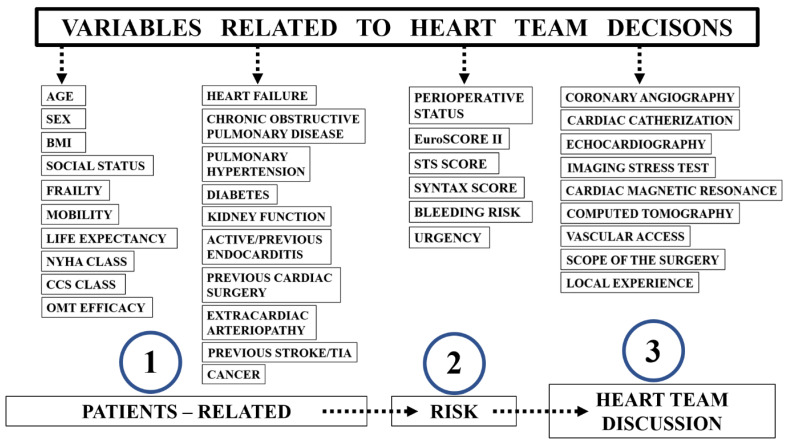
The variables associated with Heart Team decision-making process. NYHA—New York Heart Association, CCS—Canadian Cardiovascular Society, OMT—optimal medical therapy, EuroSCORE II—European System for Cardiac Operative Risk Evaluation, STS—Society of Thoracic Surgeons, SYNTAX—Synergy Between PCI with Taxus and Cardiac Surgery.

**Figure 2 jpm-12-00705-f002:**
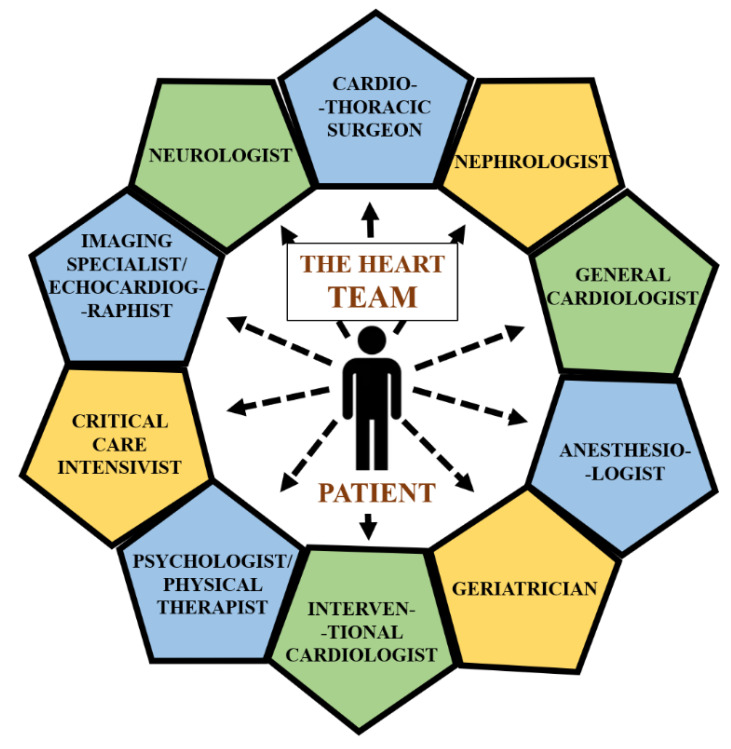
The HT specialists cooperation.

**Table 1 jpm-12-00705-t001:** Heart Team for myocardial revascularization.

Study Type	Clinical Characteristics	Results	Ref. No.
prospective, randomized	1800 patients with 3-VD or/and LMS: CABG—897, PCI—903 Follow-up: 12 months	Rates of primary—MACCE (overall mortality, stroke, MI, repeat revascularization) at 12 months were significantly higher in the PCI group (17.8%, vs. 12.4% for CABG) mostly due to an increased rate of repeat revascularization (13.5% vs. 5.9% for CABG). At 12 months, the rates of death and MI were similar between the two groups, while rates of stroke were significantly higher in CABG-patients (2.2%, vs. 0.6% with PCI).	[10], Serruys PW, et al.
retrospective analysis of prospective, randomized trial	1095 patients with 3-VD: CABG—549, PCI—546 Follow-up: 5 years	The rate of MACCE (overall mortality, stroke, MI, repeat revascularization) was significantly higher in PCI as compared with CABG-patients (37.5 vs. 24.2%). PCI vs. CABG resulted in significantly higher rates of the composite of death/stroke/MI (22.0 vs. 14.0%), all-cause death (14.6 vs. 9.2%), MI (9.2 vs. 4.0%), and repeat revascularization (25.4 vs. 12.6%). Rates of stroke were similar between groups at 5 years (3.0 vs. 3.5%).	[11], Head SJ, et al.
prospective, randomized	223 patients with de novo 3-VD or LM disease Separate HTs randomized to assess the CAD with either coronary CTA or CA.	HT compliance in the assessment of patients’ qualification for PCI or CABG procedures (the primary endpoint) was found to be very high—approximately 93%. An almost complete agreement between the two teams was demonstrated.	[12], Collet C, et al.
prospective, randomized	223 patients with 3-VD or LM disease 2 HTs to decide between CABG and PCI FFR analysis in 196 patients FFR_CT_ available for 1030 lesions	By noninvasive evaluation with FFR_CT_, the HT changed decisions for 7% of patients and modified the selection of vessels for revascularization in 12% in comparison with a coronary CTA assessment alone. For individuals assessed by coronary CTA, FFR_CT_ reduced the number of cases with hemodynamically significant 3-VD from 92.3% to 78.8%.	[13], Andreini D, et al.
prospective, nonrandomized	454 patients with de novo 3-VD without LMS compared with 315 patients from the pre-defined SYNTAX PCI group and 334 patients from the pre-defined SYNTAX CABG cohort. Follow-up: 5 years	The SYNTAX II strategy of incorporating both clinical and anatomical variables into HT decisions to guide myocardial revascularization was associated with improved 5-year clinical outcomes as compared with the SYNTAX trial, which evaluated anatomic factors only. At 5 years, MACCE (composite of all-cause death, stroke, any MI and any revascularization) occurred in 21.5% of SYNTAX II patients, which was significantly lower than the 36.4% MACCE rate in the SYNTAX PCI group. MACCE outcomes at 5 years among patients in SYNTAX II and predefined patients in the SYNTAX I CABG cohort were similar.	[14], Banning AP, et al.
retrospective	3408 catheterizations with a first diagnosis of CAD 1527 patients had first PCI Follow-up: 15 years	During follow-up of firstly PCI—patients (Kaplan–Meier analysis), CABG occurred in 15% of patients, PCI in 37% and diagnostic catheterization in 65%; mortality of any course was 51%. Mortalities were similar in 1-VD and in a population matched for age and sex, but mortality was significantly higher in firstly-PCI patients with MVD.	[15], Bonzel T, et al.
retrospective	209 patients with isolated MVD: CABG—141, PCI—59, OMT—9 Impact of hierarchy on multidisciplinary HT recommendations.	The hierarchy of the participating cardiologists and cardiac surgeons significantly impacts treatment strategies of a multidisciplinary HT. This impact did not attenuate after several years of HT interactions.	[16], Abdulrahman M, et al.
prospective	366 patients with LMS, 2-VD, 3-VD or clinical equipoise: CABG—102, PCI—127, OMT—137 Follow-up: 3 years	OMT was associated with a 4.5-fold increased risk of overall mortality compared with CABG and PCI over the 3-year period. No significant difference in overall survival at 3 years between CABG and PCI was observed.	[17], Patterson T, et al.
retrospective	960 patients with CAD—69.4%—simple CAD, 30.6%—complex CAD Median (IQR) follow-up: 4.6 (4.2–5.0) years	The 5-year mortality rates were: 16.4% for 1- or 2-VD (with proximal LAD), 15.7% for 1- or 2-VD (with non-proximal LAD), 17.1% for 3-VD, 3.4% for isolated LM or with 1-VD and 26.9% for LM with 2- or 3-VD.	[18], Dominigues CT, et al.
prospective	166 high-risk patients with complex CAD: CABG—49, PCI—79, OMT—34, hybrid therapy—1 Follow-up: 3 years	Among 129 patients who underwent revascularization (CABG or PCI) in-hospital and 30-days mortality was 3.9% and 4.8%. The 30-day unplanned rehospitalization rate was 16.4% for PCI, 22.4% for CABG and 17.6% for OMT-patients.	[19], Young MN, et al.
prospective	186 patients with MVD: 93—HT approach, 93—control group	63% vs. 23 % of patients were referred to CABG after HT discussion as compared with control group. HT discussion led to a significant delay to PCI, while delay to CABG was not affected.	[20], Kezerle L, et al.
retrospective	234 patients with MVD originally treated as recommended by interventional cardiologists (2012–2014) compared with blinded HT treatment recommendations (2017–2018)	The treatment proposed by HT showed a 30% inconsistency with the original treatment administered by the interventional cardiologists. Different treatment was recommended by the HT for 22% of patients who received CABG, 45% of patients who received PCI and 40% of patients who received medical therapy. HT members indicated statistically insignificant, but numeric bias toward the procedure of their specialty.	[21], Tsang MB, et al. Comment: [22], Blankenship JC, et al.
retrospective	1286 patients with 3-VD or/and LMS: CABG—356, PCI—679, OMT—251 Mean (SD) follow-up: 37 (14) months	In-hospital mortality did not significantly differ between treatment strategies. CABG and PCI were found to be significantly superior to OMT for primary endpoint (MACCE—overall mortality, stroke, MI, repeat/need for revascularization) and secondary endpoints (overall mortality, CV death, stroke, disabling stroke, MI, repeat/need for revascularization). For interventional strategies—CABG was associated with reduced rates of MACCE and repeat revascularization, while the superiority of PCI for stroke and disabling stroke was observed.	[23], Jonik S, et al.

1-VD—one-vessel disease, 2-VD—two-vessel disease, 3-VD—three-vessel disease, LMS—left main stenosis, CABG—coronary artery bypass grafting, PCI—percutaneous coronary intervention, OMT—optimal medical therapy, MACCE—major adverse cardiac and cerebrovascular events, MI—myocardial infarction, CAD—coronary artery disease, MVD—multivessel disease, CTA—computed-tomography angiogram, CA—conventional angiography, HT—Heart Team, FFR—fractional flow reserve, FFR_CT_—fractional flow reserve form computed-tomography, IQR—interquartile range, LAD—left anterior descending, SYNTAX—Synergy Between PCI with Taxus and Cardiac Surgery, CV—cardiovascular.

**Table 2 jpm-12-00705-t002:** Heart Team for aortic stenosis.

Study Type	Clinical Characteristics	Results	Ref. No.
prospective	163 high-risk patients with symptomatic AS: SAVR—35, TAVI—73, OMT/PTAV—55 Median (IQR) follow-up: 38 (12–42) months for SAVR, 25 (12–40) months for TAVI, 32 (18–41) months for OMT/PTAV	30-days overall mortality, CV death and stroke did not significantly differ between groups, whereas patients from SAVR group had statistically the highest 30-days incidence of life-threatening bleeding and new onset of AF. TAVI and SAVR was significantly superior to OMT/PTAV for all-cause mortality and CV death and nonsignificantly superior to OMT/PTAV for repeat hospitalizations for CV cause at 1 year. At 1 year: stroke/TIA and PPI were nonsignificantly more frequent in TAVI-group as compared with SAVR or OMT/PTAV, whereas in SAVR-group new onset of AF with the highest incidence was observed.	[31], Dubois C, et al.
retrospective	487 patients with severe AS: SAVR—392, TAVI—60, OMT—35 Median (IQR) follow-up: 3.5 (1.87–3.53) years	Very poor prognosis for OMT-group with only 57.1 and 25.7% surviving to 1 and 3 years, respectively. Survival after TAVI was lower but did not significantly differ from survival after isolated SAVR (88.3% vs. 92.6% at 1 year and 71.7% vs. 86.8% at 3 years, respectively), although TAVR-patients were older and with higher risk.	[32], Thyregod HGH, et al.
prospective	405 high-risk patients with AS: SAVR—98, TAVI—188, OMT/PTAV—116 Median follow-up: 12 months	30-days overall mortality and CV death was the most frequent in OMT/PTAV group. TAVI and SAVR was significantly superior to OMT/PTAV for all-cause mortality and CV death at 1 year. No differences in stroke/TIA and rehospitalization for CV cause between groups after 30 days and at 1 year were observed. With the highest incidence: life-threatening bleeding at 30 days, PPI and new onset of AF after 30-days and at 1 year in SAVR-group; and major vascular complications in TAVI-group after 30 days and at 1 year were observed.	[33], Bakelants E, et al.
retrospective	243 patients with severe AS: SAVR—26, TAVI—200, OMT—17 Mean (SD) follow-up: 2.0 (1.4) years	Survival outcomes after TAVI and SAVR were similar with each other and similar to the age-matched general population. Both TAVI and SAVR-patients had significantly increased survival as compared with OMT-group at 1 and 2 years.	[34], Rea CW, et al.
prospective	286 patients with AS: SAVR—53, TAVR—210, OMT—23 Median (IQR) follow-up: 18 (11–26) months	In-hospital: mortality, strokes and PPI did not significantly differ between SAVR and TAVR groups. For interventional strategies, TAVR was associated with an increased in-hospital major vascular complications, whereas in SAVR-patients significantly higher incidence of in-hospital: bleeding complications, AKI and new onset of AF were observed. 1- and 2-year all-cause mortality and CV mortality were significantly increased in OMT-group as compared with interventional strategies (SAVR or TAVR). 1- and 2-year all-cause mortality and CV mortality did not significantly differ between SAVR and TAVR.	[35], Tirado-Conte G, et al.
retrospective	482 patients with severe AS: SAVR—85, TAVR—318, OMT—79 Median follow-up: 866 days	Interventional strategies (SAVR or TAVR) was found to be significantly superior to OMT for primary (all-cause mortality, non-fatal disabling strokes and non-fatal rehospitalizations for AS) and all secondary endpoints. For interventional strategies, TAVR was associated with significantly reduced risk of AKI, new onset of AF and major bleeding, whereas in SAVR-patients significantly reduced incidence of major vascular complications and need for PPI were observed.	[36], Jonik S, et al.

AS—aortic stenosis, SAVR—surgical aortic valve replacement, TAVI—transcatheter aortic valve implantation, OMT—optimal medical therapy, PTAV—percutaneous transluminal aortic valvuloplasty, IQR—interquartile range, CV—cardiovascular, AF—atrial fibrillation, TIA—transient ischemic attack, PPI—permanent pacemaker implantation, SD—standard deviation, TAVR—transcatheter aortic valve replacement, AKI—acute kidney injury.

**Table 3 jpm-12-00705-t003:** Heart Team for mitral regurgitation.

Study Type	Clinical Characteristics	Results	Ref. No.
prospective	158 patients with MV pathology with or without concomitant cardiac disesase: Surgery—67 (MVR or MVP; isolated or concomitant), percutaneous—20 (MC or MVA), OMT—71 30-days mortality and MACCE An estimated (Kaplan-Meier) overall survival with median follow-up: 450 days	30-days mortality: surgery—4.4%, OMT—4.2%, percutaneous—0.0%. 30-days MACCE (mortality, MI, reoperation for failure or surgical repair, stroke, renal failure, infection, sepsis): surgery—16.0%, percutaneous—5.0%. 450-days overall survival: beneficial long-term outcomes for surgically treated patients and very poor prognosis for OMT-group (25.4 % overall mortality).	[40], Heuts S, et al.
retrospective	400 patients with MR: MVR—36, MVP—185, MC—179 Mean (SD) follow-up: 32.2 (17.6) months	No significant difference in in-hospital mortality between MVR, MVP and MC. MVP-patients with significantly higher 4-year survival and fewer combined endpoints (all-cause mortality, CV rehospitalization and MV reintervention) as compared with MVR and MC groups.	[41], Külling M, et al.
retrospective	1145 patients with MV disesase: 641—discussed by dedicated mitral HT (surgery—289, transcatheter—101, OMT—251); 504—discussed by general HT (surgery—285, MC—7, OMT—212) Median (IQR) follow-up: 41.1 (22.8–60.0) months	No significant difference in 30-day mortality between patients discussed by dedicated mitral HT and general HT. Rate of 1-year mortality significantly reduced and 5-year survival probability significantly increased for patients discussed by dedicated mitral HT as compared with general HT.	[42], Sardari Nia P, et al.
retrospective	157 patients with severe MR: MVR—46, MC—58, OMT—53 Mean (SD) follow-up: 29 (15) months	All-cause mortality, CV death, nonfatal MI, nonfatal stroke, nonfatal hospitalizations for HF and CV events/one patient significantly the most frequent in OMT-group. No significant difference between MVR and MC for all-cause mortality, CV death, nonfatal MI, nonfatal stroke, nonfatal hospitalizations for HF and CV events/one patient. No significant difference in in-hospital mortality between MVR and MC.	[43], Jonik S, et al.

HT—Heart Team, MV—mitral valve, MVR—mitral valve replacement, MVP—mitral valve repair, MC—MitraClip, MVA—mitral valve annuloplasty, OMT—optimal medical therapy, MACCE—major adverse cardiac or cerebrovascular event, MI—myocardial infarction, SD—standard deviation, IQR—interquartile range, CV—cardiovascular, HF—heart failure.

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
