# Peer review of "An Individualized Approach of Multidisciplinary Heart Team for Myocardial Revascularization and Valvular Heart Disease—State of Art"

_jpm, 2022, doi:10.3390/jpm12050705_

Round 1

Reviewer 1 Report

The authors demonstrated the important review suggesting that multidisciplinary heart team approach for management of heart disease patients. Although the concept of the review is of important, there are several major concerns that should be addressed. Specific comments are followings.

  1. Regarding heart disease patients, management of patients’ adherence is a very important factor for improving prognosis or QOL. If present, please mention the reports on adherence in the area of coronary artery disease or valvular heart disease.
  2. There are many references that compare treatments. For mainly suggesting the importance of the heart team approach, please select the papers on the team approach more appropriately.

Author Response

Dear

We greatly appreciate you having reviewed our manuscript.

Below we presented our responses to your comments.

  1. About what you asked: “Regarding heart disease patients, management of patients’ adherence is a very important factor for improving prognosis or QOL. If present, please mention the reports on adherence in the area of coronary artery disease or valvular heart disease.” – We agree with your opinion – for patients with coronary artery disease (CAD) and valvular heart disease (VHD) management of patients’ adherence is a very important factor for improving prognosis or QOL. Unfortunately, for papers regarding HT concept, these kind of data are very scarce. We searched once more all cited articles attached to our review and only in references [10] and [14] we found some mentions regarding patients’ adherence to physicians recommendations.

   We have added in the text:

1) For reference no. 10 – Section: Heart Team for myocardial revascularization. Page 5-6:

" The researchers reported that the use of antiplatelet drugs was high in the PCI arm (with 71.1% receiving a thienopyridine at 12 months). Additionally, both the use of statin, angiotensin-converting enzyme inhibitors or angiotensin II receptor blockers and calcium channel blockers was significantly higher after the study procedure in PCI group. Highly effective dual–antiplatelet and statin therapy may prevent thromboembolic incidents, hence the lower rates of stroke in patients undergoing PCI. " and " and adherence to drugs use in optimal medical therapy (OMT) group ".

2) For reference no. 14 – Section: Heart Team for myocardial revascularization. Page 7:

" In addition, an optimized medical therapy was a part of SYNTAX II strategy. An increased use of statins at 5 years following revascularization (83 % in SYNTAX-I PCI vs 88 % in SYNTAX II; p=0.055) may be responsible for some of the improved outcomes of patients from SYNTAX II cohort. Furthermore, pre-procedural loading dosing of statins may be associated with meaningful decrease in periprocedural rates of MI in SYNTAX II. Although, the aspirin and dual antiplatelet therapy (DAPT) recommendations were significantly more frequent in the SYNTAX II group at discharge,  rates of ADP antagonist prescription at 5 years were much higher in SYNTAX-I PCI. Probably, this fact can be explained by lower rates of repeat revascularization/MI and utilization of new generation of drug-eluting stents (DES) with less dependence of DAPT in SYNTAX II cohort. Other CV medications were similarly used among SYNTAX II and SYNTAX-I PCI. "

Of course, we agree that a poor coverage of this issue is a limitation to this review, so we have highlighted it in the section “Limitations”:

Page 37: " Additionally, for patients with CAD and VHD, adherence to physician recommendations and regular drugs usage is very important factors of future prognosis and quality of life. Unfortunately, these parameters are very difficult to measure and often remain a matter of mutual trust between the doctor and the patient. Also, among the articles regarding HT that we have cited in this manuscript, this issue is overlooked and very rarely raised [10], [14]. So, a proper qualitative assessment of adherence to medical recommendations is still an unexplored issue and adds to the limitations of our review. "

------------------------------------------------------------------------------------------------------------

  1. About what you asked: “There are many references that compare treatments. For mainly suggesting the importance of the heart team approach, please select the papers on the team approach more appropriately.” – We agree with your opinion – unfortunately data specifically related to Heart Team approach is poorly mentioned in the available literature. We found only two of them and we have added in the text:

    Section: Heart Team for myocardial revascularization. Page 9:

" Tsang MB, et al. reported very interesting results from a study of 234 patients with MVD comparing the treatment originally implemented by interventional cardiologists (2012 – 2014) with recommendations proposed by members of 8 blinded HTs (2017 – 2018). Between the original decisions of the interventional cardiologists and the results of the HT consultations, a different decisions occurred in nearly one-third of the cases. HT members indicated statistically insignificant, but numeric bias toward the procedure of their specialty. Overall, as the choice of the treatment strategy is regarding, there were no statistically significant differences between interventional cardiologists and HT members for CABG (P=0.62) or PCI (P=0.15), while OMT was less frequently recommended originally by interventional cardiologist than by the HT members (P=0.04). ([21]; and comment - [22]) "

And in Table 1. Heart Team for myocardial revascularization. Page 16-17:

retrospective

234 patients with MVD originally treated as recommended by interventional cardiologists (2012-2014) compared with blinded HT treatment recommendations (2017 – 2018)

·       The treatment proposed by HT showed a 30% inconsistency with the original treatment administered by the interventional cardiologists.

·       Different treatment was recommended by the HT for 22% of patients who received CABG, 45% of patients who received PCI and 40% of patients who received medical therapy.

·       HT members indicated statistically insignificant, but numeric bias toward the procedure of their specialty.

[21], Tsang MB, et al.

Comment: [22], Blankenship JC, et al.

Also, we have added references to it: Section: References – Page 42-43:

  1. Tsang MB, Schwalm JD, Gandhi S, Sibbald MG, Gafni A, Mercuri M, Salehian O, Lamy A, Pericak D, Jolly S, Sheth T, Ainsworth C, Velianou J, Valettas N, Mehta S, Pinilla N, Yanagawa B, Zhang L, Chu V, Parry D, Whitlock R, Dyub A, Cybulsky I, Semelhago L, Ioannou K, Hameed A, Wright D, Mulji A, Darvish-Kazem S, Gupta N, Alshatti A, Natarajan MK. Comparison of Heart Team vs Interventional Cardiologist Recommendations for the Treatment of Patients With Multivessel Coronary Artery Disease. JAMA Netw Open. 2020; 3(8): e2012749. DOI: 10.1001/jamanetworkopen.2020.12749.
  2. Blankenship JC, Mercado N. Treatment Recommendations for Patients With Multivessel Coronary Artery Disease-There Is No "I" in Heart Team, But Is the Heart Team Better Than the I? JAMA Netw Open. 2020; 3(8): e2013098. DOI: 10.1001/jamanetworkopen.2020.13098.

            Of course, we agree that a poor coverage of this issue is a limitation to this review, so we have highlighted it in the section “Limitations”:

Page 36-37: As we noted in the introduction, currently the main limitation of the HT concept still remains the lack of well-founded, step by step – planned RCTs comparing the long-term outcomes of patients treated with and without the HT approach. To date, evidence of the advantages of implementing multidisciplinary decision-making has been derived mainly from experts opinion and observational studies without a comparator. Unfortunately, in most of the large studies we referenced in this manuscript, such as SYNTAX, SYNTAX II, PARTNER, PARTNER 1, NOTION, PARTNER 2, SURTAVI, EVEREST II or COAPT [10], [14], [24], [27], [28], [29], [30], [37], [38] the main theme is head-to-head comparison of various treatment options for myocardial revascularization, AS or MR, rather than an importance or specific role of HT in the management of patients with these diseases. Therefore, performing of well-designed RCTs with hard clinical endpoints remains one of the most important perspectives for a future HT concept.

We believe you are satisfied with our answer. Thank you for taking the time for review this article.

With best regards

Szymon Jonik, MD

1st Department of Cardiology

Medical University of Warsaw,

Banacha 1a Street
01-267 Warsaw, Poland
e-mail: [email protected]
tel.: + 48 22 599 19 58
fax.: + 48 22 599 19 57

Reviewer 2 Report

Jonik et al. presented an extensive review of multidisciplinary heart team (MHT) role and various results on their decision in the management of patients with coronary artery disease or valvular heart disease. Although there are limited well-designed studies on the role of MHT in the era of numerous treatment options for CAD and VHD with increasing complexity of our patients who need optimal treatment for their multiple comorbidities, I would suggest several concerns on this comprehensive manuscript.

Major.

  1. The authors used both terms, MHT and HT, throughout the manuscript. Is there any particular reason or difference between the terms which made the authors use similar terms separately? Unifying key terms throughout the paper may deliver a more clear message.

  1. The major limitation of this topic would be a shortage of rigorous well-designed studies investigating the role or advantages of multidisciplinary heart team (MHT). Many of literature the authors provided in this paper are dealing with mostly the efficacy of various treatment options for heart disease rather than the role of MHT in the management of the patients. Especially in original articles of EVEREST II and COAPT trial the main theme is head-to-head comparison of treatment options for mitral regurgitation, rather than an importance or specific role of MHT in the management of mitral regurgitation. I don’t think these kinds of papers present insightful evidence for the main topic of this review. Please consider these issues as limitation of the paper.

  1. I would suggest that the authors may consider and include more relevant references focused on the role of MHT in the management of patients with CAD or VHD. The authors may consider papers of JAMA Netw Open 2020 Aug 3;3(8):e2012749. doi: 10.1001/jamanetworkopen.2020.12749. and JAMA Network Open. 2020;3(8):e2013098. doi:10.1001/jamanetworkopen.2020.13098.

  1. It would more ordered form of this review article if the aortic stenosis part goes before the mitral regurgitation part, as the MHT for AS treatment (TAVR vs. SAVR) may take up much more part of their role than that for MR (TEER vs. surgical correction or medical treatment).

Minor.

  1. Page 7, please provide full term when use abbreviation for the first time (OMT).

  1. Page 6, line 6 from bottom, please put closing parenthesis between “revascularization” and “occurred”.

  1. Page 8, line 7 from bottom, put the p value in the parenthesis.

  1. Page 18, line 6, please put the p value (P=0.002) at the end of the sentence in the parenthesis.

  1. Page 18, line 5 from bottom, please put the p value in the parenthesis.

  1. Page 22, line 12, write full name first, then abbreviation (PTAV).

  1. There is no need to abbreviate any terms to just use once for all throughout the manuscript. In page 24, line 4 (AF), lines 5-6 (PPI), page 29, line 3 (IVUS and CTO), these abbreviations look unnecessary.

Author Response

Dear

We greatly appreciate you having reviewed our manuscript.

Below we presented our responses to your comments.

  1. About your comment no.1: “ The authors used both terms, MHT and HT, throughout the manuscript. Is there any particular reason or difference between the terms which made the authors use similar terms separately? Unifying key terms throughout the paper may deliver a more clear message. “

            Our answer: We have standarized the nomenclature - all 'MHT' abbreviations we changed into 'HT' abbreviations to make the text more understandable.

  1. About your comment no.2: “The major limitation of this topic would be a shortage of rigorous well-designed studies investigating the role or advantages of multidisciplinary heart team (MHT). Many of literature the authors provided in this paper are dealing with mostly the efficacy of various treatment options for heart disease rather than the role of MHT in the management of the patients. Especially in original articles of EVEREST II and COAPT trial the main theme is head-to-head comparison of treatment options for mitral regurgitation, rather than an importance or specific role of MHT in the management of mitral regurgitation. I don’t think these kinds of papers present insightful evidence for the main topic of this review. Please consider these issues as limitation of the paper. “

            Our answer: Section: Limitations – Page 36-37: As we noted in the introduction, currently the main limitation of the HT concept still remains the lack of well-founded, step by step – planned RCTs comparing the long-term outcomes of patients treated with and without the HT approach. To date, evidence of the advantages of implementing multidisciplinary decision-making has been derived mainly from experts opinion and observational studies without a comparator. Unfortunately, in most of the large studies we referenced in this manuscript, such as SYNTAX, SYNTAX II, PARTNER, PARTNER 1, NOTION, PARTNER 2, SURTAVI, EVEREST II or COAPT [10], [14], [24], [27], [28], [29], [30], [37], [38] the main theme is head-to-head comparison of various treatment options for myocardial revascularization, AS or MR, rather than an importance or specific role of HT in the management of patients with these diseases. Therefore, performing of well-designed RCTs with hard clinical endpoints remains one of the most important perspectives for a future HT concept.

  1. About you comment no.3: “ I would suggest that the authors may consider and include more relevant references focused on the role of MHT in the management of patients with CAD or VHD. The authors may consider papers of JAMA Netw Open 2020 Aug 3;3(8):e2012749. doi: 10.1001/jamanetworkopen.2020.12749. and JAMA Network Open. 2020;3(8):e2013098. doi:10.1001/jamanetworkopen.2020.13098. “

Our answer: We have added references you asked:

Section: Heart Team for myocardial revascularization. Page 9:

" Tsang MB, et al. reported very interesting results from a study of 234 patients with MVD comparing the treatment originally implemented by interventional cardiologists (2012 – 2014) with recommendations proposed by members of 8 blinded HTs (2017 – 2018). Between the original decisions of the interventional cardiologists and the results of the HT consultations, a different decisions occurred in nearly one-third of the cases. HT members indicated statistically insignificant, but numeric bias toward the procedure of their specialty. Overall, as the choice of the treatment strategy is regarding, there were no statistically significant differences between interventional cardiologists and HT members for CABG (P=0.62) or PCI (P=0.15), while OMT was less frequently recommended originally by interventional cardiologist than by the HT members (P=0.04). ([21]; and comment - [22]) "

And in Table 1. Heart Team for myocardial revascularization. Page 16-17:

retrospective

234 patients with MVD originally treated as recommended by interventional cardiologists (2012-2014) compared with blinded HT treatment recommendations (2017 – 2018)

·       The treatment proposed by HT showed a 30% inconsistency with the original treatment administered by the interventional cardiologists.

·       Different treatment was recommended by the HT for 22% of patients who received CABG, 45% of patients who received PCI and 40% of patients who received medical therapy.

·       HT members indicated statistically insignificant, but numeric bias toward the procedure of their specialty.

[21], Tsang MB, et al.

Comment: [22], Blankenship JC, et al.

Also, we have added references to it: Section: References – Page 42-43:

  1. Tsang MB, Schwalm JD, Gandhi S, Sibbald MG, Gafni A, Mercuri M, Salehian O, Lamy A, Pericak D, Jolly S, Sheth T, Ainsworth C, Velianou J, Valettas N, Mehta S, Pinilla N, Yanagawa B, Zhang L, Chu V, Parry D, Whitlock R, Dyub A, Cybulsky I, Semelhago L, Ioannou K, Hameed A, Wright D, Mulji A, Darvish-Kazem S, Gupta N, Alshatti A, Natarajan MK. Comparison of Heart Team vs Interventional Cardiologist Recommendations for the Treatment of Patients With Multivessel Coronary Artery Disease. JAMA Netw Open. 2020; 3(8): e2012749. DOI: 10.1001/jamanetworkopen.2020.12749.
  2. Blankenship JC, Mercado N. Treatment Recommendations for Patients With Multivessel Coronary Artery Disease-There Is No "I" in Heart Team, But Is the Heart Team Better Than the I? JAMA Netw Open. 2020; 3(8): e2013098. DOI: 10.1001/jamanetworkopen.2020.13098.
  3. About you comment no.4: “It would more ordered form of this review article if the aortic stenosis part goes before the mitral regurgitation part, as the MHT for AS treatment (TAVR vs. SAVR) may take up much more part of their role than that for MR (TEER vs. surgical correction or medical treatment). “

            Our answer: We have changed it and the ‘aortic stenosis’ part and also adequate references are now before ‘mitral regurgitation’ part.

  1. About you comment no.5: “Page 7, please provide full term when use abbreviation for the first time (OMT).”

            Our answer: We have added it. The first time the ‘OMT’ abbreviation was used is on Page 5-6 and in this place the full term ‘optimal medical therapy’ was added.

  1. About you comment no.6: “Page 6, line 6 from bottom, please put closing parenthesis between “revascularization” and “occurred”. “

            Our answer: This was corrected on page 7, line 4 from the top.

  1. About you comment no.7: “Page 8, line 7 from bottom, put the p value in the parenthesis. “

            Our answer: This was corrected on page 9, line 15 from the top.

  1. About you comment no.8: “Page 18, line 6, please put the p value (P=0.002) at the end of the sentence in the parenthesis. “

            Our answer: This was corrected on page 26, line 7 from the bottom.

  1. About you comment no.9: “Page 18, line 5 from bottom, please put the p value in the parenthesis. “

            Our answer: This was corrected on page 27, line 3 from the top.

  1. About you comment no.10: “Page 22, line 12, write full name first, then abbreviation (PTAV). “

            Our answer: This was corrected on page 19, line 12 from the top.

  1. About you comment no.11: “There is no need to abbreviate any terms to just use once for all throughout the manuscript. In page 24, line 4 (AF), lines 5-6 (PPI), page 29, line 3 (IVUS and CTO), these abbreviations look unnecessary. “

            Our answer: This was corrected on page 21, line 2 and 3 from the top and on page 38, line 7 from the top.

We believe you are satisfied with our answer. Thank you for taking the time for review this article.

With best regards

Szymon Jonik, MD

1st Department of Cardiology

Medical University of Warsaw,

Banacha 1a Street
01-267 Warsaw, Poland
e-mail: [email protected]
tel.: + 48 22 599 19 58
fax.: + 48 22 599 19 57

Round 2

Reviewer 1 Report

The authors appropriately respond to my comments.